# Scaling Up Ecovillagers' Lifestyles Can Help to Decarbonise Europe

**Franziska Wiest** [1,*], **M. Gabriela Gamarra Scavone** [2], **Maya Tsuboya Newell** [3], **Ilona M. Otto** [4] and **Andrew K. Ringsmuth** [4,5]

1   Faculty of Environmental, Regional and Educational Sciences, University of Graz, 8010 Graz, Austria
2   Faculty of Geoscience, Utrecht University, 3584 CB Utrecht, The Netherlands
3   Graduate School for International Development and Cooperation, Hiroshima University, Hiroshima 739-8511, Japan
4   Wegener Center for Climate and Global Change, University of Graz, Brandhofgasse 5, 8010 Graz, Austria
5   Complexity Science Hub Vienna, 1080 Vienna, Austria
*   Correspondence: franziska.maria.wiest@gmail.com

**Abstract:** Decarbonisation is an essential response to the threat of climate change. To achieve Europe's net-zero 2050 climate targets, radical technological and social changes are required. Lifestyle changes for reducing greenhouse gas (GHG) emissions are an important component of complex systemic transformation. The typical behaviour of inhabitants in ecovillages is potentially more conducive to sustainable lifestyles than the current European standard lifestyle. This study explores the potential of ecovillagers' lifestyles to contribute to decarbonisation using the Multilevel Perspective (MLP) theoretical framework. The research data were obtained through the model tool EUCalc and an online survey of 73 ecovillage residents in 24 European countries. The results indicate that current ecovillagers' lifestyles, regarding home, consumption, diet, and mobility, would continue to produce 40% fewer emissions per capita than the standard European lifestyle by 2050. The study identifies which ecovillage behaviours would produce the largest reductions in per-capita $CO_2$eq emissions if adopted by society more broadly.

**Keywords:** climate change; demand-side solutions; lifestyle change; decarbonisation; ecovillage; EUCalc; Europe; GHG emissions; transformation pathway; transition; multilevel perspective

## 1. Introduction

Europe requires new integrative strategies to support a more sustainable future, with climate change mitigation as a high priority. The Paris Agreement [1] set targets for the greenhouse gas (GHG) emissions budget that would limit global warming to below 1.5 °C or 2 °C by the end of the century [2]. Yet, these targets cannot be fulfilled solely through technological and policy measures [3]. The Sixth Assessment Report of the IPCC (AR6) recognises that demand-side solutions with associated changes in lifestyles can support near-term climate mitigation [4,5]. It has become evident that common European behaviours contribute significantly to an individual's carbon footprint. However, further investigation is required into which demand-side behavioural changes could lead to climate mitigation pathways [5]. A recent study showed that a shift towards sustainable lifestyles in Europe is a pivotal pathway in all future scenarios that may influence both biodiversity and sustainable agriculture practices [6]. Identifying key lifestyle practices that entail GHG-saving potential can provide important leeway for Europe to meet its mitigation targets [7]. These can be further explored by looking at alternative life models and practices.

An alternative model for the design and reorganisation of living is explored within intentional communities, such as ecovillages [8,9]. An ecovillage is a community that is consciously designed through locally owned participatory processes in all dimensions of sustainability (social, culture, ecology, and economy) to regenerate social and natural

environments [10,11]. Some central principles of ecovillage living are to share spaces and to live self-sufficiently through socially and ecologically conscious lifestyle choices [11].

Previous studies have found that inhabitants in ecovillages, also known as ecovillagers, have a lower environmental impact and stronger social support when compared with the broader society [12–14]. Most prior research on ecovillages has been through case studies, and few have used large sample sizes. Wagner et al. reviewed 59 qualitative individual case studies on ecovillages from 2000 to 2012, showing that ecovillages placed high importance on the social context of lifestyle [15]. Further research by Simon et al. also investigated three ecovillages in Germany and compared this to three reference families (German national average), finding that these ecovillages had a significantly lower ecological footprint [16]. However, no research so far has explicitly examined the decarbonisation potential of wider adoption of lifestyles that are currently practised in ecovillages.

This paper aims to analyse to what extent ecovillagers' lifestyles and behaviours can contribute to decarbonising Europe if adopted by the broader society. Our study goes beyond prior work by collecting a larger sample to quantify the ecological impacts of ecovillagers' lifestyles and identifies which behaviours can be adopted by society more broadly. The GHG emissions of ecovillagers' behaviours are calculated using the online tool European Calculator (EUCalc) and compared with emissions of the standard European citizen [17]. EUCalc is a model for representing behavioural and technological changes and their impact on GHG emission dynamics [17]. The online tool allows users to build hypothetical pathways to a net-zero carbon future by creating different scenarios. The GHG emissions associated with various degrees of behavioural and technological change can be calculated and visualised. Finally, we explore which aspects of ecovillagers' lifestyles in the areas of mobility, housing, diet, and consumption hold the largest potential for contributing to decarbonisation if adopted by wider society.

The paper is structured as follows: first, we introduce ecovillages' historical development and their role in systemic transformation. Second, we describe our methods of data collection and analysis with EUCalc. Finally, we present and discuss key findings.

## 2. Ecovillages as Niche Innovation

### 2.1. The History of Ecovillages

Alternative lifestyles and intentional communities date back to the Roman Empire, where ancient communal traditions existed [18]. Intentional communities centred on religious values until early in the 19th century, when political communities began to arise. In the 1990s, the global movement of ecovillages arose as a new way of living in Europe, North America, and Oceania [13]. This emergence has been linked to the postmodernist social movement of the time, characterised by fragmenting and differentiating from the mainstream. People with similar ideas began to contact each other, form groups, and withdraw from mainstream society [18].

Ecovillages began to move beyond mainstream society within the rural and urban areas as an alternative model for the design and reorganization of living in the twenty-first century [8]. These examples of ecovillages existed in both developed and developing countries, with a range of different forms and foci (e.g., spiritual/ecological, urban eco-architectural, permaculture farming communities, and traditional ecological living) [13]. As the number of ecovillages continued to grow, the Global Ecovillage Network (GEN) was established to interconnect them.

The GEN [10] defines an ecovillage as a community that has the following three core practices:

1. Being rooted in local participatory processes.
2. Integrating social, cultural, economic, and ecological dimensions in a whole systems approach to sustainability.
3. Actively restoring and regenerating their social and natural environments.

Furthermore, each ecovillage has its ethos and principles that ground its community and overall vision of ecovillage living. The number of registered ecovillages worldwide

in 2021 was 900, with 358 in Europe within the GEN [10]. Given the large diversity across ecovillages, there remains limited demographic data about the entire global population of ecovillagers.

### 2.2. Multi-Level Perspective as a Framework for Transition

To analyse the transition potential in the broader context of society, we need to consider the ecovillage within a framework that links systems across scales. The multi-level perspective (MLP) framework is one approach among others for conceptualising transformations to sustainability in sociotechnical systems [19]. It combines ideas from evolutionary economics, sociology of innovation, and neo-institutional theory to conceptualise overall dynamic patterns in system transformation [20]. Contrary to a top-down approach, the MLP allows the inclusion of different actors and highlights the role of individuals in the transition (bottom-up approach). It comprises a nested hierarchy of regimes embedded within landscapes and niches within regimes (Figure 1). Although there are three different levels, the system is characterized by being dynamic. Thus, transformations result from the interaction between processes at different levels [19].

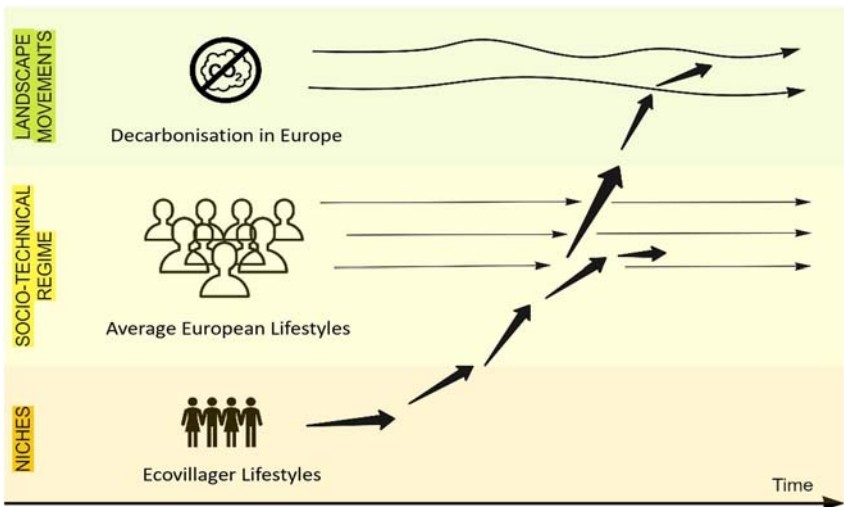

**Figure 1.** MLP on transitions (adapted from [21]).

The MLP has previously been applied to, for example, mobility system transitions in Europe. Moradi and Vagnoni combined a literature review with stakeholder interviews conducting qualitative data analysis to identify the main mobility regimes in Italy and the dynamics of low-carbon mobility transitions, thus revealing current pathways and most likely pathways in the scope of 2030 targets [22]. Köhler et al. combined qualitative case studies of mobility niches in the Netherlands with quantitative simulations to develop transition pathways, including both behavioural and technological change [23]. Such studies have demonstrated the merits of the MLP by going beyond studies of single technologies, which dominate the literature on environmental innovation, and instead emphasising multidimensionality and structural change across interacting system levels [20]. In light of such a successful combination of case studies and simulation under the MLP framework, we apply a similar approach in the present study.

As lifestyle is a big factor in decarbonisation, we look at ecovillages (niche innovation) to understand possible transitions in the future (Table 1). According to [21], first, niche innovations build up internal momentum. Second, changes at the landscape level create pressure on the regime, and third, destabilisation of the regime creates windows of opportunity for niche innovations. In terms of this framework, the imperative for decarbonisation on the landscape level is putting pressure on the socio-technical regime: the 'normal' way that people live. Under this pressure and instability, opportunities arise for ecovillagers' lifestyles to gain momentum and be adopted by the broader society.

**Table 1.** Geels MLP framework applied to the research (own compilation, source: [21]).

| Levels | Application to the Research | Characteristics (Geels) |
|---|---|---|
| Landscape *Macro-level* | Decarbonisation in Europe | Slow changing external factors, providing gradients for the trajectories |
| Socio-technical regime *Meso-level* | Standard European lifestyles | Stability of existing technological development and occurrence of trajectories |
| Niches *Micro-level* | Ecovillagers' Lifestyles in Europe | Generation and development of radical innovations |

## 3. Materials and Methods

### 3.1. EUCalc

EUCalc is a model for representing the behavioural and technological changes that shows the impact on GHG emission dynamics from 1990 until 2050 at the scale of Europe [24]. EUCalc [17] was developed with scientific and societal actors within the European Union's Horizon 2020 research and innovation programme. The online tool allows users to model energy, resources, production, and food systems at the EU level under pre-defined but adjustable levels of ambition with regard to technological deployment and consumption behaviour to build hypothetical pathways to a net-zero carbon future [25]. Through the user interface—the Transition Pathways Explorer—different emission scenarios can be created. The GHG emissions from different levels of behavioural and technological change can be calculated and directly visualised [19].

When evaluating decarbonising potentials in research, it is common to calculate GHG emissions with one-sided optimisation models [24]. The EUCalc, however, provides a more in-depth analysis due to its cross-sectoral approach [26]. This model represents GHG emissions dynamics until 2050 and can be applied for delineating emission and sustainable transformation pathways at the scale of Europe. GHG reductions are calculated by adjusting certain parameters in different factors, such as *Key behaviours*, *Technology and Fuel*, *Resources and Land use* and *Boundary conditions* [7,17]. This helps with understanding the impact of choices such as lifestyle changes and technological options and the key implications for policy planning [17]. Accordingly, EUCalc closely mirrors the MLP in its multilevel structure, quantifying interdependences between lifestyles at the micro level, technological regimes at the meso-level and the macro-level political imperative for rapid decarbonisation.

EUCalc uses two approaches: (1) define a calculation sequence based on material, energy and emissions drivers and (2) set a range of ambition levels on the drivers. The drivers are categorised within different scopes; the drivers of energy within the scopes are called levers. Instead of calculating the user's optimal pathway, this model allows the user to set each lever flexibly to an ambition level (AL) on a scale rising from one to four (Figure 2). ALs reflect the degree of ambitiousness of each lever regarding GHG emission reduction by considering the demand for products or activities [17]. The AL four (AL4) results in the highest contribution to GHG emissions reduction until 2050. For each AL, EUCalc provides a merit value (threshold) at which a specific AL can be achieved. An example for AL4 of the lever *Car occupancy* is the threshold of 2.6 people per car (Figure 3). In addition, the user can choose the climate scenario on which he/she wants to base his/her calculation [7]. Since this research focuses on lifestyles, the scopes of the category *Key behaviours* were considered: Travel, Homes, Diet and Consumption (Figure 2). EUCalc includes three further categories, namely *Technology and fuels*, *Resources and land use* and *Boundary condition* in pathway calculations. Yet, these are not considered lifestyle and behaviour specific, so they were omitted in this research. Instead, the EU-Reference data from EUCalc, in alignment with [27], was adopted for all calculations.

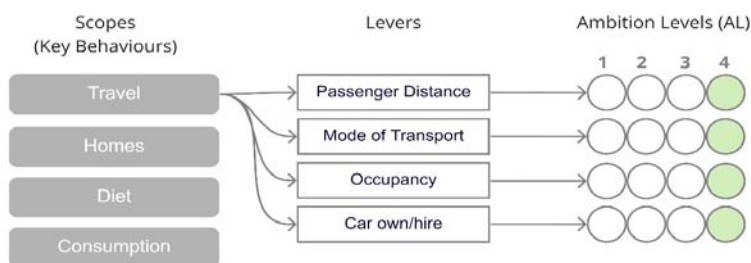

**Figure 2.** Scopes, levers and ALs of the *Key Behaviours* category in EUCalc.

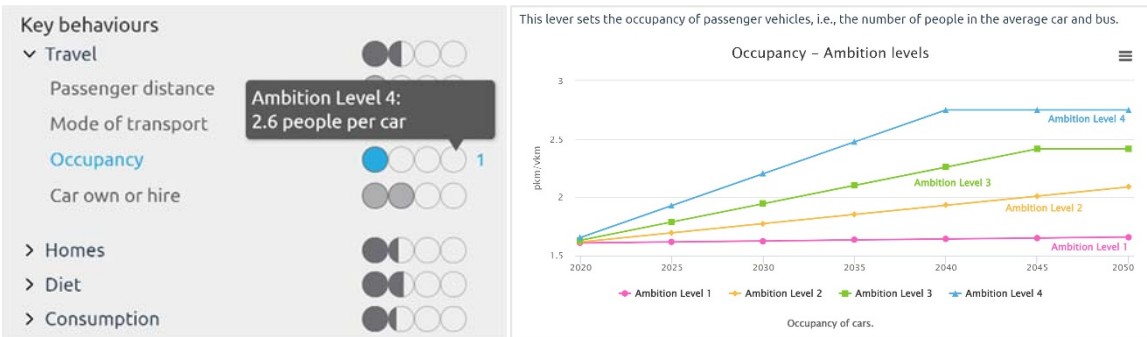

**Figure 3.** The lever 'Occupancy' (scope *Travel*) is an example to illustrate how the definition of AL4 was obtained and then translated into an exact numerical threshold [17].

### *3.2. Survey Process*

3.2.1. Survey Construction from the Key Behaviour Levers of the EUCalc

A survey was created with 28 questions based on the EUCalc interface of the *key behaviours* section (*Travel*, *Homes*, *Diet* and *Consumption*) and the definitions of respective levers. This survey aimed to create questions for each lever to determine whether the respondent meets or surpasses the threshold to comply with AL4, the highest ambition level defined by EUCalc. The interface of EUCalc allows setting each lever of the scopes manually to different ALs. Hovering over the AL, a definition is displayed, naming the respective threshold that needs to be reached for the respective AL. To illustrate this procedure with an example, Figure 3 displays how the definition of AL4 for the lever 'Occupancy' of the scope *Travel* was obtained and then translated into an exact numerical threshold of 2.6 persons per car (Figure 3).

These thresholds then served as a basis for the creation of the survey questions. According to this procedure, a survey question was derived for each lever of all four scopes. However, some levers were not precisely defined as a quantifiable threshold by EUCalc and were thus disregarded in the survey. Out of 15 levers, three had to be excluded. Furthermore, the order of the questions was slightly changed from the original order of EUCalc, to create a more concise and logical order of the survey questions to the respondents. For each question, answer options were given, out of which at least one option translates into meeting the defined threshold of the respective AL4. Hence, questions and answers were formulated in a way that provides similar options instead of fewer contrasting options to avoid nudging participants. The language of the survey was English.

Based on the example of Figure 3, Table 2 shows the process of creating the survey question and answer options based on the AL4 threshold obtained from EUCalc [17]. Depending on the answer chosen by a respondent, an answer counted as 'Meeting AL4' or 'Not meeting AL4'. If more than 50% of the respondents met AL4 in a respective lever, the lever was considered as Meeting AL4 in the final results.

**Table 2.** The lever 'Occupancy' (scope *Travel*) is an example to illustrate the process of creating the survey question and answer options based on the AL4 threshold obtained from EUCalc [17].

| Scope | Lever | AL 4 Threshold | Survey Question | Answer Options | Meeting AL4 |
|---|---|---|---|---|---|
| **Travel** | *Occupancy* | 2.6 people per car | If you travel by car, how many people are in the car most of the time, including yourself? | (a) I barely travel by car.<br>(b) At least 2 people.<br>(c) At least 3 people.<br>(d) Only myself. | Yes<br>Yes<br>Yes<br>No |

Note: Full set of questions and answers is available upon request.

### 3.2.2. Data Collection

The lifestyle data were collected directly from ecovillage members via an online survey. The structure of the survey is based on the scopes and levers of EUCalc [17]. To reach a wide range of participants in age, nationality and profession, a large sample size was required. The pool of available communities was identified through the GEN Europe [10] database, a repository of information on existing and forming ecovillage communities around the world. Through the GEN database as a seed source, the snowball sampling technique was applied, and the ecovillages were contacted and invited to participate, as well as asked to forward the survey to other members and/or ecovillages. To recruit respondents relating to our survey for ecovillagers residing in Europe, we also used the social networking site Facebook between May 2021 and June 2021. The six closed Facebook groups used for attaining respondents were: "Eco-Village and Intentional Community Discussion Group", "Vegan Ecovillage in Europe", "Global Ecovillage Network group", "Ecovillages Europe & World Tour", "Communauté autonome france (projet ecolieux eco village permaculture)" and "Ecovillage Connection Europe". Messages with the link to the survey were sent out directly via email and Facebook to 158 ecovillages in 29 European countries.

### 3.2.3. Sample of Responses Obtained

In total, 73 residents of ecovillages in 24 countries participated in the survey (Figure 4). As shown in Figure 4b, Germany, Austria and the Netherlands had the highest numbers of respondents. This does not entirely reflect the distribution of ecovillages in Europe, given that Spain has the highest number (Figure 4a).

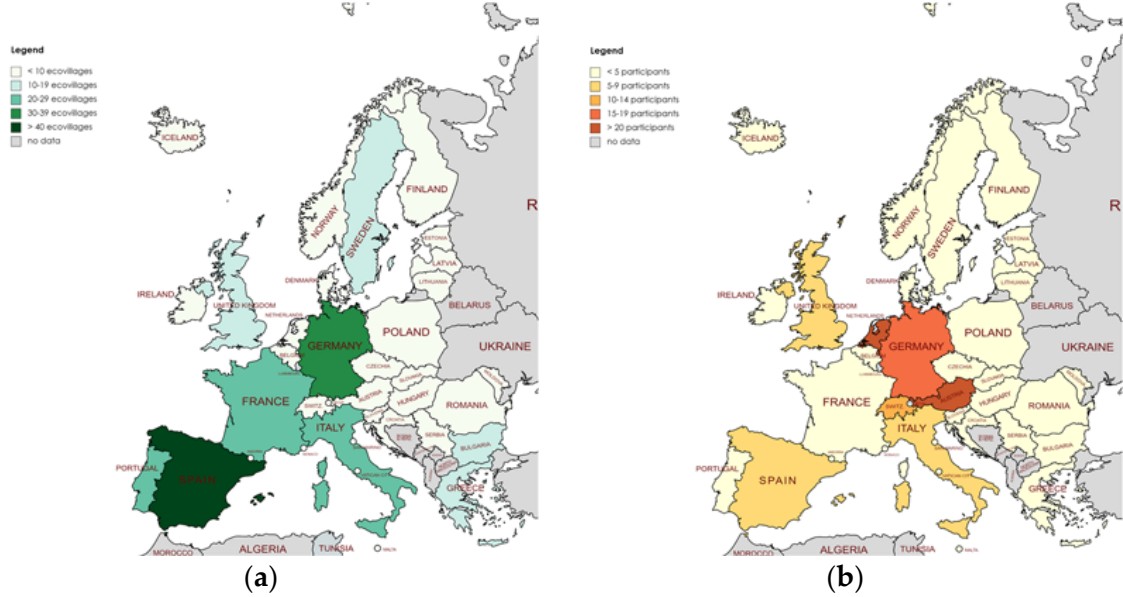

(**a**) (**b**)

**Figure 4.** Distribution of (**a**) ecovillages in Europe and (**b**) survey participants per European country (source: [10], own data).

All of the questions were single-choice and mandatory. The responses were typically given by one ecovillage member, acting as a representative of their community. Although it would have been ideal for every ecovillage member to reply, our goal was to include as many communities as possible, so we opted to limit the respondent burden and increase the total response rate. As expected, the completion rate was quite low; several reasons may account for this. Some communities, during the inquiry process, openly did not support the institutional system of society and hence refused to participate in a survey from an institution such as a university. To give the survey a less formal frame, the invitation text, sent by email, was written using intentional, personal language and supported by photos of the conducting researchers.

The contacted ecovillage members were kindly asked to participate and, if possible, to forward the survey link to other members. The researchers pointed out that participation was completely voluntary and that they could be contacted at any time regarding concerns about data security or the clarity of the questions. Nevertheless, most of the contacted ecovillages did not reply to emails and did not participate. The contacted ecovillage members who did participate in the survey mostly did so without any further comment. Only very few replied to emails, but some even showed interest in the survey, asking for further details of the research or requesting to be notified about the results afterwards.

### *3.3. Data Analysis*

### 3.3.1. Analysis of the Survey

The data analysis aimed to investigate whether ecovillagers' lifestyles can help to reduce GHG emissions in Europe. The EUCalc model was used to calculate the overall GHG emission reduction potential of the lifestyles in ecovillages relative to the European lifestyle aligned with [27]. For the calculation, the AL of each lever was set to either AL4 or the European reference value, which is pre-set in the interface of EUCalc. Whether a lever was set to AL4 was determined by the following process.

As explained in Section 3.2, questions provided answer options 'Meeting AL4' and options 'Not meeting AL4'. Depending on the answer chosen by a respondent, an answer counted as 'Meeting AL4' or 'Not meeting AL4'. If more than 50% of the respondents met AL4 in a respective lever, the lever was considered as 'Meeting AL4' in the final calculation. If fewer than 50% of the respondents' answers did not comply with 'Meeting AL4', the lever was set to the lever aligned with the European reference scenario in [27] (see Figure 5).

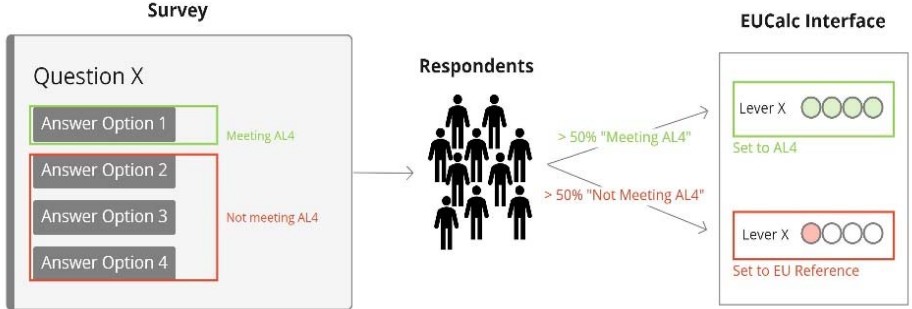

**Figure 5.** Data analysis example.

### 3.3.2. Analysis of the GHG Emissions

The next step was to calculate and compare the transition pathways until the year 2050 for different lifestyles. We compared the ecovillagers' lifestyles based on the AL's from the survey results and another for the European lifestyle aligned with [27]. The analysis considered the main sectoral assumption and outputs of the EU-Reference as detailed in [27] with data from EUCalc. Furthermore, a third transition pathway was calculated for the lifestyle with the maximum GHG abatement potential. This lifestyle was calculated with EUCalc considering that all *key behaviours*' scopes met AL4 and was created for comparison purposes.

To assess the relevance of this carbon reduction to the broader context, we considered the total carbon budget in scenario analysis. The total carbon budget is defined by the IPCC [28] and EUCalc [17] as the GHG emissions that can still be emitted worldwide to have two out of three chances of limiting global warming by 1.5 °C or 2 °C by the end of this century. Thus, two factors are considered important for mitigating climate change. First, limiting the temperature rise to either 2 °C or 1.5 °C relative to preindustrial levels by the year 2050 through a reduction in $CO_2$eq emissions [21,22]. Second, consider the factor of the 'fair share' of global $CO_2$eq emissions. Here, two options are considered: (1) every person in the world should be allowed to emit the same amount of GHG ('per capita'), or (2) Europeans should emit less GHG due to their above-average GDP compared to people from less developed countries ('capability') [17].

Considering the two factors, EUCalc presented four different scenarios with different European shares of $CO_2$eq emissions depending on the parameters selected (Figure 6). The four scenarios were considered to assess the results of the GHG emissions reduction in this study (Section 4.2).

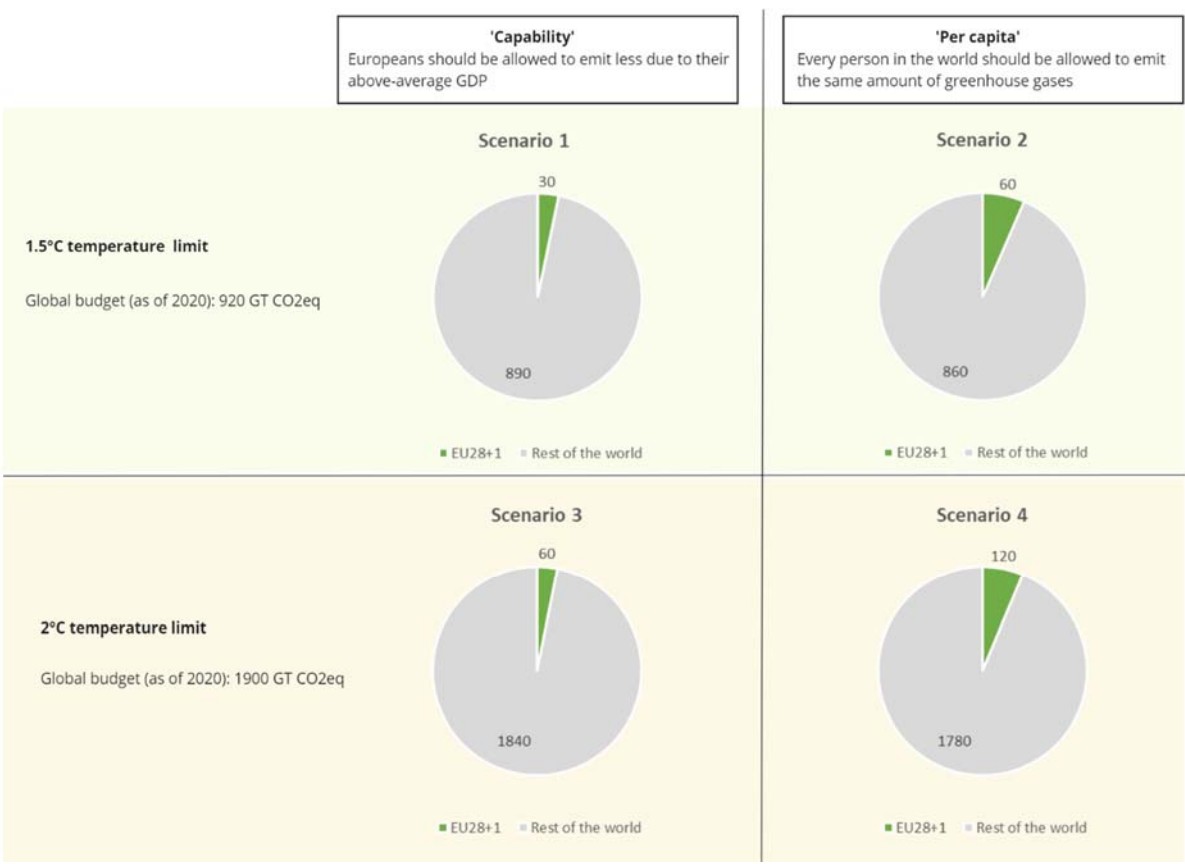

**Figure 6.** The European share of GHG emission (in dark green) for four different scenarios until 2050 (source: [17]).

## 4. Results

The goal was to obtain results that reflect the standard European ecovillagers' lifestyle. This study included respondents from a wide range of residence countries, ages and occupations. The results of the survey are based on the sample size of N = 73. Considering residence country and age, quite a diverse distribution was achieved (see Figure 3). Notably, 72% graduated from university, and 36% were self-employed. Only 11% were farmers or dedicated to agriculture-related work, and 7% were specialist workers such as artisans or craftspeople.

*4.1. Ambition Levels (ALs) of Lifestyle Levers in Ecovillages*

Lifestyles in ecovillages were quantified and measured based on scopes, respective levers and ALs of EUCalc. Table 3 depicts an overview of the most relevant results with one example of each lever.

Scope *Travel*, Lever *Occupancy*: The EU reference complies with AL1, which is an average of 1.6 people per car [19]. According to the survey, 86% of ecovillagers achieve compliance with AL4.

Scope *Homes*, Lever *Appliances owned*: The EU reference meets AL2, which claims that the standard European household aligned with [27] owns 1.3 computers [19]. In this case, only 18% of the survey participants complied with AL4, while 84% did not meet AL4.

**Table 3.** Overview of relevant survey results (source: own survey data, Supplementary Materials).

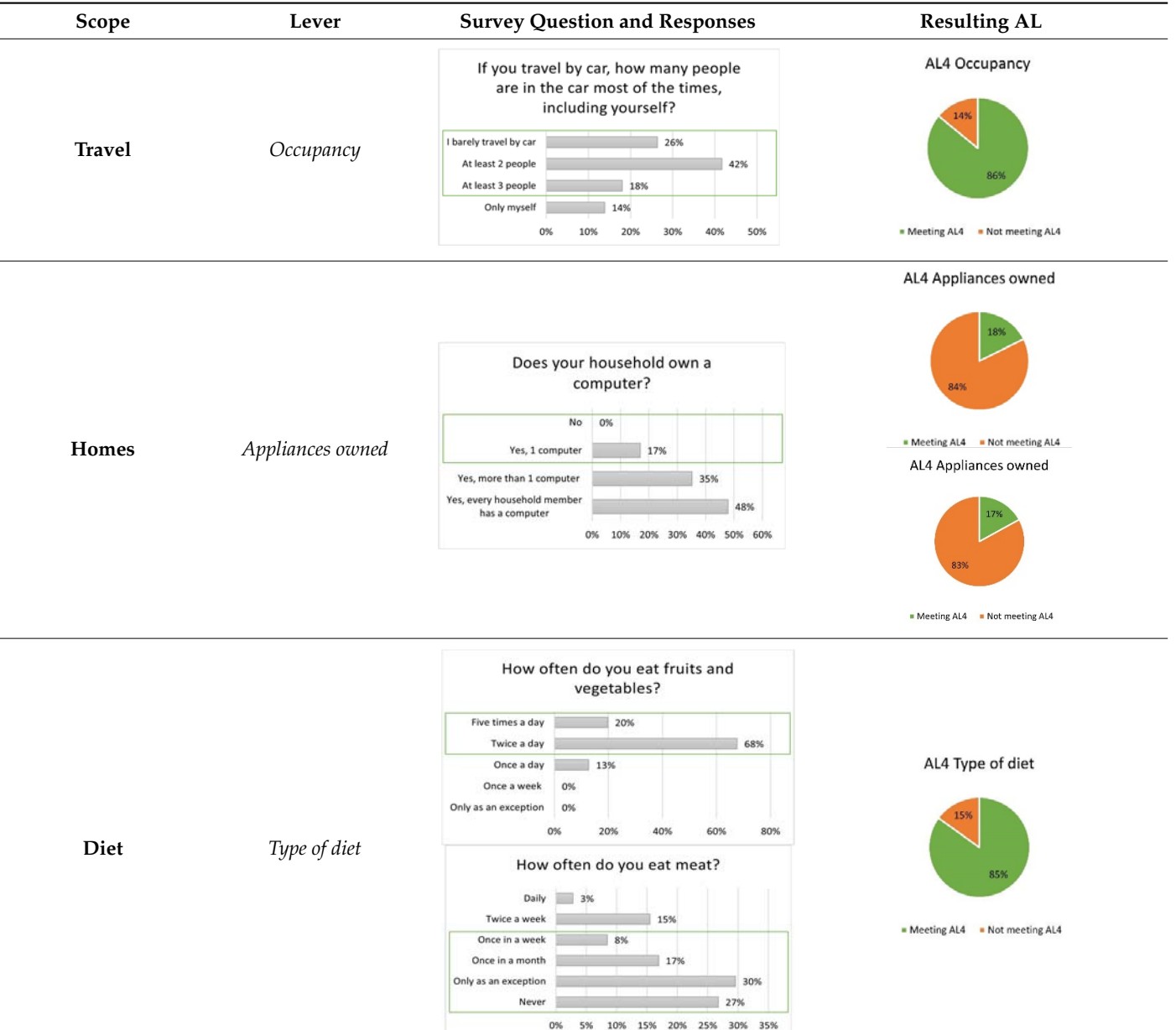

| Scope | Lever | Survey Question and Responses | Resulting AL |
|-------|-------|-------------------------------|--------------|
| **Travel** | *Occupancy* | | |
| **Homes** | *Appliances owned* | | |
| **Diet** | *Type of diet* | | |

**Table 3.** *Cont.*

| Scope | Lever | Survey Question and Responses | Resulting AL |
|-------|-------|-------------------------------|--------------|
| **Consumption** | *Food waste* | 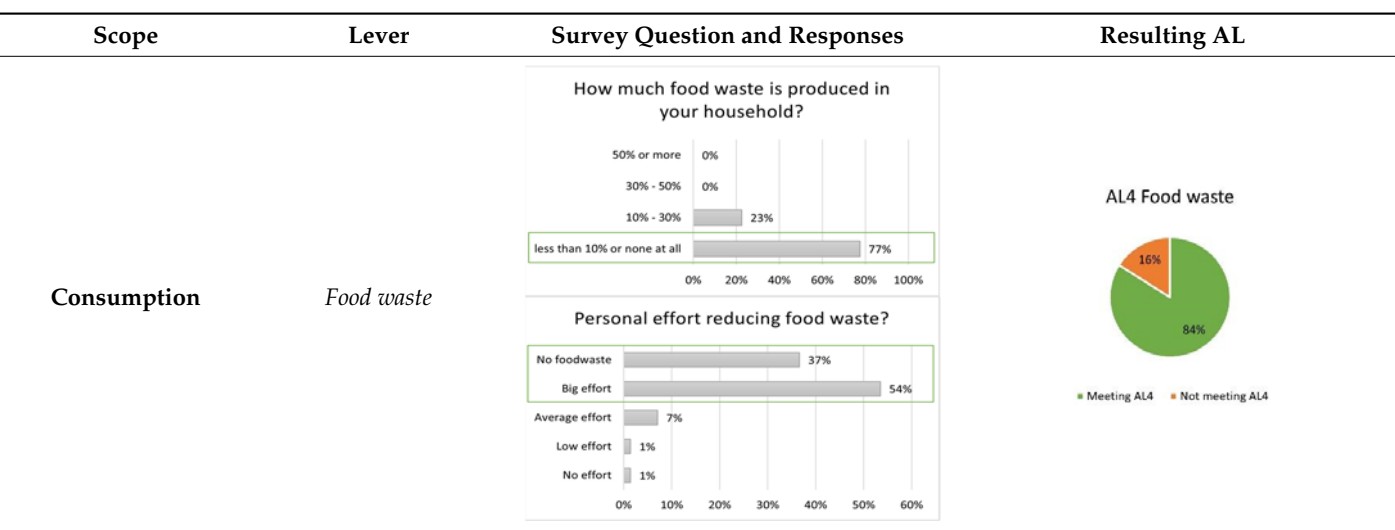 | |

Scope *Diet*, Lever *Type of Diet*: The standard European can be scaled at AL1 regarding their type of diet, accounting for a meat calorie intake of 331 kcal per person per day [17]. In contrast, 85% of the ecovillagers who participated in the survey chose to reduce their meat intake drastically, as AL4 is only 92 kcal per person per day [17]. This was measured in frequency in the survey, as this is a more accessible way of asking the question. It was assumed that if a person eats meat only once per week, 92 kcal per day cannot be exceeded.

Scope *Consumption*, Lever *Food waste:* On average, 521 food kcal per person per day go is wasted in Europe [17]. The survey results found that ecovillagers put high efforts into reducing food waste or do not produce any food waste at all. Out of the two questions targeting food waste reduction efforts and the amount of food waste produced, results showed that 80% of ecovillagers met AL4.

### 4.2. Relevant Aspects of Eco-Living for GHG Emissions Reduction

To identify the most relevant findings of obtained results, the data are analysed in a second explorative step. Thus, it is possible to analyse the potential of ecovillagers' lifestyles to reduce GHG emissions and to identify which lifestyle aspects of eco-living can be considered relevant and transferable to the broader society. As explained in Section 3, EUCalc is used for the GHG calculation based on the survey results. Table 4 provides an overview of levers that were set to AL4 and which remained with the EU reference value. It also depicts the levers that were excluded and which were difficult to measure. Furthermore, it indicates the percentage of participants that met, and did not meet, AL4.

**Table 4.** ALs of levers used for GHG calculation in EUCalc (source: own data).

| Scope | Lever | AL | Percentage of 'Meeting AL4' |
|-------|-------|----|-----------------------------|
| **Travel** | *Passenger distance* | **1** | 46% * |
| | *Mode of transport* | **4** | 85% |
| | *Occupancy* | **4** | 86% |
| | *The car owns or hire* | **4** | 77% |
| **Homes** | *Living space per person* | **1** | 48% |
| | *Percentage of cooled living space* | **4** | 97% |
| | *Space cooling & heating* | **2** | * |
| | *Appliances owned* | **2** | 18% |
| | *Appliance use* | **2** | 16% |
| **Diet** | *Calories consumed* | **2** | * |
| | *Type of diet* | **4** | 85% |

**Table 4.** *Cont.*

| Scope | Lever | AL | Percentage of 'Meeting AL4' |
|---|---|---|---|
| **Consumption** | *Use of paper and packaging* | 4 | 59% |
| | *Appliance retirement timing* | 4 | 88% |
| | *Food waste* | 4 | 84% * |
| | *Freight distance* | 1 | * |

Note: * Levers are difficult to quantify in the survey; this is clarified in the discussion.

Levers Meeting AL4

Table 4 provides an overview of the AL reached for each lever based on the survey's results. The most relevant scopes and levers—those for which more than 50% of respondents met AL4—are shown in green.

Figure 7 depicts the four scopes and respective levers as follows: green signifies higher compliance (≥50%), and blue, lower compliance (<50%) according to AL4 percentages (Table 4). The levers in green are strongly relevant due to remarkably high percentages (≥75%). The levers that were difficult to survey and, therefore, hard to quantify were excluded and appeared as grey fields. The levers in blue did not meet AL4 and were left unchanged from the standard European lifestyle aligned with [27], provided by EUCalc.

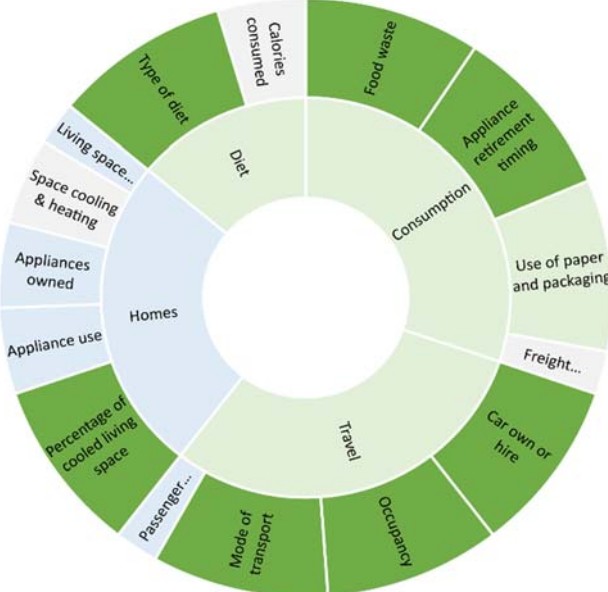

**Figure 7.** Relevance of levers according to AL4 percentage (source: survey data, Supplementary Materials).

*4.3. GHG Emissions Comparison among Lifestyles and Scenarios*

The transition pathways until the year 2050 were calculated for three lifestyles (Figure 8). The first is the European lifestyle, which was aligned with [27] and considered the main sectoral assumption and outputs of the EU-Reference as detailed in [27] and can be selected with data from EUCalc [17]. Second, the ecovillagers' lifestyle is based on the ALs in Table 4 from our survey results. Finally, the lifestyle with the maximum GHG abatement potential considering that all *key behaviours'* scopes meet AL4 and can be selected with data from EUCalc [17].

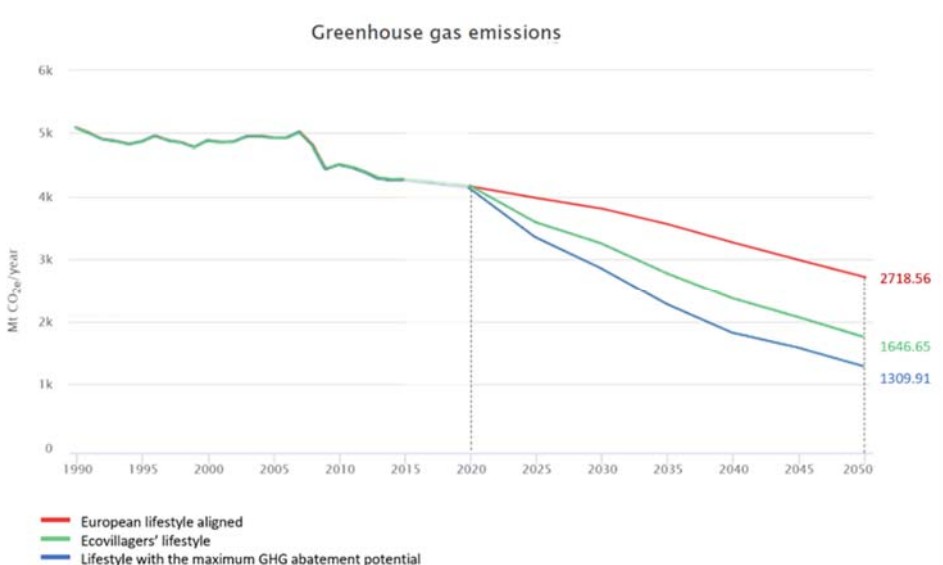

**Figure 8.** GHG emission standard European lifestyle (red) [27], ecovillagers' lifestyle (green) and lifestyle with the maximum GHG abatement potential (blue) in [Mt $CO_2$eq/y] (source: [17], own calculation).

Our research shows that by 2050, the GHG emissions for the ecovillagers' lifestyle are 1647 Gt $CO_2$eq/year and the European lifestyle 2719 Gt $CO_2$eq/year. This shows that the ecovillagers' lifestyle produces 40% (1072 Gt $CO_2$eq) less per year than the European lifestyle aligned with [27] if adopted by the broader society. Yet, the lifestyle with the maximum GHG abatement potential (1310 Gt) is 20% lower than the ecovillagers' lifestyle and 52% lower than the European lifestyle aligned with [27].

Considering the adoption of the respective lifestyles by the whole EU until 2050, EU-Calc calculated 138 Gt cumulative $CO_2$eq for the ecovillage lifestyle and 207 Gt cumulative $CO_2$eq for the standard European lifestyle aligned with [27]. This represents an overall reduction of 33.3% (69 Gt). Considering the four scenarios described in Figure 6, we assessed the results for the three lifestyles (Figure 9).

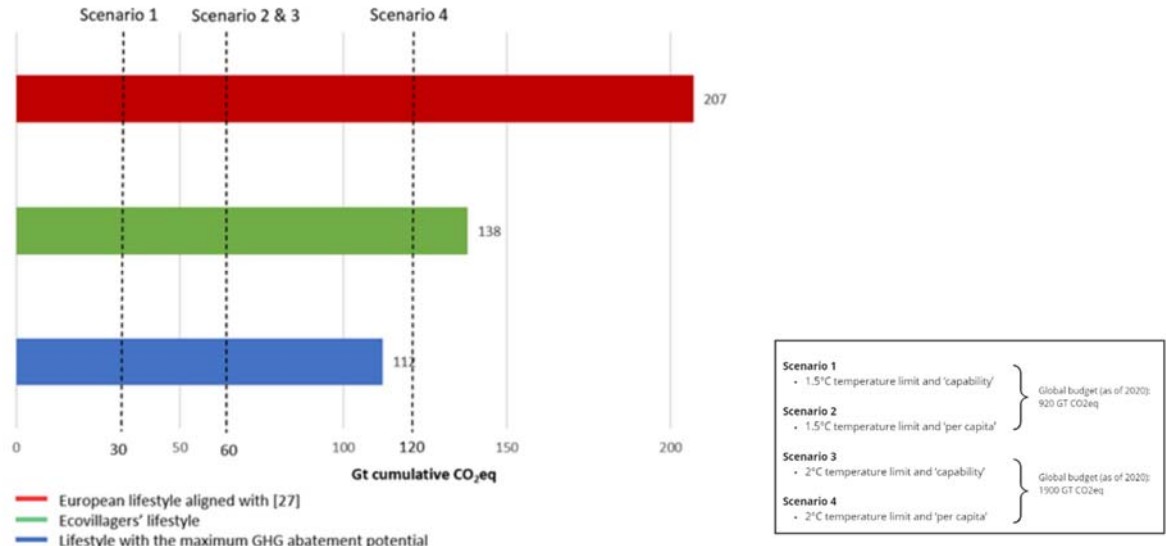

**Figure 9.** European budget [Gt cumulative $CO_2$eq] for each lifestyle based on the four scenarios. (Source: [17], own data).

Both the standard European lifestyle (207 Gt $CO_2$eq) and the ecovillagers' lifestyle (138 Gt $CO_2$eq) would exceed the European GHG budget by 2050 for the four scenarios.

However, the ecovillagers' lifestyle, considering *Home*, *Consumption*, *Diet* and *Travel*, would continue to produce significantly lower GHG emissions per capita than the standard European lifestyle by 2050. These results suggest that behavioural changes in the European lifestyle [27] would have a big impact on GHG emissions. Furthermore, the lifestyle with the maximum GHG abatement potential would produce 112 Gt cumulative $CO_2$eq, which is 20% less than the ecovillagers' lifestyle and 46% less than the standard European lifestyle [27]. This is the only lifestyle that meets the carbon budget for scenario 4.

## 5. Discussion

### 5.1. Transferability of the Results

Focusing on demand-side solutions broadens the climate mitigation approach beyond the supply side to include solutions that improve health and well-being whilst reducing energy and material throughput [29]. Although it is unrealistic to assume that all EU citizens can live in ecovillages, the results provide insight into mitigation potentials through varied lifestyle approaches on the demand-side and reveal the areas that require a shift towards low-carbon infrastructure and service provisions.

As suggested by the MLP framework [19–21], ecovillage lifestyles can be seen as niche innovations, and some of their current lifestyle patterns might diffuse to the main socio-technical regime and change the overall societal landscape [30]. To give an example, the currently increasing energy prices may drive increased use of public transportation, car-sharing facilities, as well as smaller living spaces, perhaps sufficiently to change the dominant lifestyle standards. Policy support, such as subsidising public transportation and lowering ticket prices, as well as encouraging the development of more dense human settlements with land use regulations and building code changes, might help to speed up the diffusion of some ecovillage lifestyles aspects. The EUCalc [17] is a useful tool that can help with understanding the potential for such scale-up and the dynamic relationship between individual choices, landscape changes, and their potential to contribute to reaching the global or regional goals of climate policy.

This study shows that there are alternative pathways to dealing with the climate crisis, and it is imperative that more research be undertaken into real-world-applicable solutions. Although ecovillages were neglected and considered little more than social experiments in the past, here they are seen to be an example of sustainable lifestyle choices with practical implications [23]. This paper's contribution to research extends the *key behaviours* outlined in EUCalc and identifies which behaviours reach the ambition of the 1.5 °C to 2 °C targets set by the Paris Agreement [1]. It also helps to extend the alternative community literature by highlighting unsustainable behaviours in typical ecovillagers' lifestyles. *Key behaviours* that achieve the highest ambition level (AL4) can be considered as behavioural changes that have the potential to substantially reduce GHG emissions and are hence relevant to broader society.

#### 5.1.1. Travel

Our results show that ecovillagers' lifestyles achieve AL4 in almost the entire *Travel* scope. Ecovillagers show that their transport choices are not centred around car ownership specifically and that alternative transport choices exist within these communities. The mode that citizens choose for travel has a large impact on energy use and emissions since fossil-fuelled cars have higher emissions per person compared to public transport [17]. Ecovillagers were found to use a range of transport modes in addition to the car, frequently choosing buses and trains. The results also revealed that although many ecovillagers do rely on a car for transport, they engage in effective carpooling strategies with at least two people in the car. The ecovillagers also share their car within the community, providing it for neighbours, friends, or other community members that need it for travel. Ecovillagers' transportation habits show a sustainable solution that directly reduces transportation costs and $CO_2$ emissions [31]. Carpooling has become a common practice among ecovillagers sharing with community members or family members when a car is needed. Descriptive

norms may have encouraged these communities to try carpooling or sharing and thus led to this behavioural practice [32].

However, although the results showed that ecovillagers use a wide range of transport and carpooling, the lever *Passenger Distance* shows no significant deviations from the standard EU citizen (Table 2). This suggests a strong potential for demand-side measures to mitigate climate change due to transportation needs [19]. Demand for personal travel is driven by the amount of time a person needs to spend travelling, given the average speed of transportation systems. The fact that about 77% of the participants are over 29 years old and 72% have a university degree could be a reason why there are no deviations in the lever *Passenger Distance* compared to the data of EUCalc and why ecovillagers may also reach AL1. This is because one of the three main activities that account for most of the time spent travelling is going to work or studying [17]. In addition, higher levels of wealth and education are associated with more time spent on leisure travel [17]. This suggests a need to consider reducing travel demands for going to work or study. This could be achieved by, for example, continuing the uptake of online study and work practices that was accelerated by the COVID-19 pandemic [33]. The travel sector is particularly difficult to decarbonise due to the investment costs of building low-emission transport systems or the slow turnover of stock and infrastructure [17].

### 5.1.2. Diet

Concerning the *Diet* scope, the results show that the ecovillagers in this survey met AL4 regarding their type of diet since more than 50% eat meat as an exception or do not include meat in their diet at all. The vegetarian and vegan diet models assume a similar amount of total food consumption. However, instead of meat, there is an increased consumption of other food categories [8]. Studies from the Institute for Prospective Technical Studies of the European Commission show that meat and dairy cause over 70% of the life cycle environmental impacts for household consumption within Europe [34]. Ecovillagers' alternative diet may act as a model for dietary change in other European households to help remain within a safe GHG budget. More than half of the ecovillages in the survey consume local produce with a few exceptions, and 35% vary according to the season. With ecovillagers consuming locally, their diets emit less GHG and contribute less to waterway eutrophication through agricultural nutrient runoff relative to importing the food [9]. The results also show a clear desire among ecovillagers to reduce food waste. More specifically, 77% of ecovillagers claimed to waste less than 10% of their food, and when asked about their effort level, 54% indicated a big effort, with 37% stating that they produce no food waste. Many ecovillages have composting sites available, which reduce damage to the environment by lowering the total amount of waste disposed [8].

### 5.1.3. Consumption

Regarding the *Consumption* scope, ecovillagers' levers met AL4 very often, consistent with their statements that respect for nature and the Earth is a central issue [35]. In addition to their low levels of food waste generation, low consumption of packaging can also be attributed to this principle. Since a lot of attention is paid to self-sufficiency, fewer packaging materials are used.

The survey distinguished consumption related to ownership and the usage of personal household appliances from other forms of consumption. Ecovillagers were found to retire their appliances beyond their expected lifetime. Rather than constantly purchasing new products, the lifespan was extended through repairing or repurchasing. This correlates with the idea of sustainable living contrary to consumerism [36].

Ecovillagers' consumption patterns regarding food waste, appliance retirement and the use of paper and packaging all met the levels needed to reach 1.5 °C. This highlights the importance of intrinsic values and the active choice of ecovillagers to avoid extensive consumption [13]. Transferring this to broader society could be translated into raising

awareness and enhancing education about the effects of consumption on GHG emissions amongst the broader society.

### 5.1.4. Homes

For the *Homes* scope, the only dimension on which ecovillagers reach AL4 is the fraction of their habitat that is cooled. Buildings consume large quantities of energy through heating, ventilation and air conditioning systems [37]. Reducing energy use in buildings is important for reducing GHG emissions, and 97% of ecovillagers were able to live in houses with energy-efficient cooling. Architecture remains a key feature in energy-saving, and ecovillagers' houses must have special features to enable this.

Although ecovillagers appear to be energy efficient in the summer months, more than 43% of the respondents have more than 80 m$^2$ of living space. The amount of residential space affects the energy use intensity of buildings and the number of raw materials needed during construction. Relative to a standard European citizen, ecovillagers have a lot of space [17]. However, as ecovillages live collectively in spaces shared with several or all members, perhaps the living space may be more loosely defined [38].

It should also be mentioned that the ecovillagers´ ownership and use of appliances appear to not reach AL4. Almost half of all respondents state that each member of their household owns a computer or laptop. Most are self-employed or employed at a company. Thus, digital technology is an essential item for many workers, including ecovillagers. The ecovillagers' use of other household appliances, such as washing machines, refrigerators, and freezers, was reportedly shared within households. As the highest percentage of ecovillagers live in households with five or more people, the potential for sharing these products is high. Regarding their ICT appliance usage, ecovillagers also reported a high screen time, similar to the standard European citizen, which is around 3 h daily [39]. The use and ownership of ICT appliances produce a high level of GHG emissions from metal resource use to the final disposal. Therefore, this lifestyle behaviour for ecovillagers is exhibited similarly to the standard European citizen due to similar personal or work commitments [40].

Considering all the results, we conclude that a broad societal shift to ecovillagers´ lifestyles alone would not limit global warming by 1.5 °C or 2 °C under the four scenarios considered. For Europe to decarbonise, it is important to also address other factors considered by the cross-sectoral approach of EUCalc, such as *Technology and Fuel*, *Resources and Land use* and *Boundary conditions*, as these have led to unsurpassable emission bottlenecks posed by geography and infrastructure regardless of lifestyle behavioural choice. For instance, ecovillagers' food consumption patterns may be influenced by easier purchasing options, and the environmental impact of technology required for work could act as an emission bottleneck.

Nevertheless, this research shows that changes in *Key behaviours* play a crucial role in meeting European targets and that the sustainable practices already in European ecovillages can be more widely adopted and scaled up across Europe.

### 5.2. Limitations of the Survey and Further Research

The survey was created within the boundaries placed by EUCalc, which is limited because the model itself does not reflect the full complexity of the real world. The researchers made judgments to combine various ALs or sector trajectories. The survey data were limited, as several communities are not registered in the GEN database or chose not to participate. Therefore, the survey underlies a selection bias for the communities most engaged in the discussion and movement about communal living. As mentioned in Section 3.2, a further limitation is our attempt to obtain household-level equivalent responses, which biased the responses to some extent. As the responses are given by individual ecovillagers, a level of subjectivity is unavoidable. Similarly conducted studies in ecovillage research have also experienced the limitation of household-level-style responses, for example [11]. This warrants attention in future studies.

The survey was based on the four key behaviour scopes established by EUCalc: *Travel*, *Homes*, *Diet* and *Consumption*. However, the order of the questions was intentionally changed to create a cohesive and clear flow of content for the respondent. Furthermore, some of the indicators were difficult to transform into questions, as values for AL4 are provided in a very specific way. For instance, the indicator Passenger distance describes the total distance travelled per year with very similar intervals indicating different ALs. It is difficult for the respondent to estimate their travel distance per year. Other indicators, such as ´Calories consumed´, were not included in the survey, as no answerable question that would produce valid results could be formulated. It should also be noted that this study was conducted during the COVID-19 pandemic, which prevented any face-to-face contact with respondents, perhaps leading to sampling bias.

We constrained our analysis to the EU level because we received responses from fewer than five participants from most member states, and this was not adequate to support statistically robust country-level analysis or comparison between countries. This prevented us from taking full advantage of the country-level resolution of EUCalc, which would be an interesting goal for future research.

Although within the *Home* scope, ecovillagers were deemed not to meet AL4, this was based only on a partial picture of home energy use; including more factors, such as electricity consumption, may change the results. There might also be other related energy issues that are connected to ecovillagers' lifestyles, such as land practices, which were not explored in this study due to the EUCalc's predefined scopes. Exploring these could be an interesting avenue for future studies.

EUCalc has proven to be a useful tool for analysing the lifestyles of ecovillagers, and we offer some reflections on its limitations, based on our experience, aiming to assist with improvements for future studies. In some indicators, the thresholds for the different ALs are given in large numbers (e.g., km travelled per year or calories consumed per year), making it difficult to translate them into a survey context which seeks to explore real-life situations. On a similar note, the scopes *Consumption* and *Homes* both include home appliances (TV, computer, etc.), which we found difficult to distinguish since one is about the usage beyond end-of-life and the other about hours used per day. Based on our study, we suggest differentiating how many electronic devices are used in a household, how many need twenty-four-hour electricity supply (such as fridges) and how many hours TV and computers are used. In addition, we suggest adding levers to the *Homes* scope: the source of electricity (e.g., grid or own supply through renewable energy) and the awareness of energy sources and consumption behaviours. Additionally, it would be helpful to extend the *Diet* scope with levers asking about the consumption of packaged or unpackaged food as well as regionally or non-regionally produced food and similar levers. In addition to the food waste considered in this scope, waste, in general, could be handled in more detail, including other ways of treating waste or the amount of waste produced in a specific time frame.

One of the key strengths of EUCalc is its visualisation tools. However, our experience revealed small points for improvement. The units could be consistent throughout the model (e.g., on the graph, $CO_2$eq is currently expressed in Mt but in the calculations in Gt). Furthermore, it would be useful to be able to calculate how much each person should emit for each one of the four scenarios along the current cumulative $CO_2$eq for the whole European population. Despite the importance of collective $CO_2$eq reductions, people as individuals should be aware of the impact of their lifestyles ($CO_2$eq per year) to be able to better manage their behaviour. Moreover, it would be helpful if EUCalc could display multiple scenarios on a single plot under a particular 2050 target.

This research has identified what lifestyle behaviours in current ecovillages meet AL4. How ecovillagers adopted these behaviours, however, was not studied. Since lifestyles rely on social and technological structures, further investigation into the structures that have catalysed the adoption and continuation of ecovillagers' sustainable behaviours would help to identify targets for policy and innovation that could encourage a broader-

scale transition to these behaviours. The role of actors such as businesses and industries, policymakers and politicians, consumers, civil society, engineers and researchers is crucial to the transition because they reproduce, maintain and transform the sociotechnical regimes in which niche innovations such as the ecovillagers' lifestyles take root. Barriers to enacting certain behaviours should also be studied, with attention to how these may differ between ecovillages and other contexts.

Many of the changes discussed are within the agency of individual citizens [41], including, for example, the choice of household appliances and modes of transportation. Information and education campaigns, as well as labels providing information about energy use and CO2 emissions related to the production of different goods and services, would help individuals to make more informed choices [5]. However, there are also limits to individual agency, and legal, policy and infrastructure changes are needed to support individuals. Subsidy programs aimed at, for example, supporting tickets for public transportation or car-sharing facilities, as well as building codes and standards, such as spaces in apartment buildings for shared washing rooms and freezers, can be recommended. Incentivising smaller living spaces in new buildings could also be helpful.

To sum it up, our results indicate a need to include more detailed lifestyle options in the integrated assessment models that are commonly used in the research on climate mitigation [41]. To give an example, the demand-side solutions for climate mitigation modelled in the recent IPCC report [42] do not include the whole range of options, such as lowering the living space and extending the time of appliance use, which have a high potential to reduce emissions if they can be scaled up. Furthermore, our results show that lifestyle aspects from the scopes *Travel*, *Homes* and *Diet*, including, for example, reducing travel by car, living in more energy efficient buildings with shared appliances and reducing meat consumption and food waste, have in particular a strong potential to reduce GHG emissions. Therefore, we recommend operationalising some of these dimensions of lifestyle changes in integrated assessment models and prioritising these in future research. However, our research also shows that there are limits to GHG emission reductions with behavioural and lifestyle changes, and they must be accompanied by infrastructure, policy and broader changes in social norms to be effective.

### 6. Conclusions

This research investigated whether ecovillager behaviours could constitute niche innovations, the widescale adoption of which could help to decarbonise Europe. The behaviours were assessed through EUCalc's ALs as a reference benchmark for decarbonisation. We found that ecovillagers' current lifestyles would continue to produce significantly lower carbon per capita than the average European lifestyle aligned with [27] by 2050.

We highlight two contributions to this research area. First, our survey provided the first large-scale Europe-wide demographic data collection from ecovillages. Second, our results suggest that widescale adoption of ecovillagers' behaviours could cut GHG emissions by up to 40% relative to the average European lifestyle, aligned with [27]. The survey highlighted the importance of *Travel*, *Homes*, *Diet* and *Consumption* practices as key indicators for reducing the carbon footprint of an individual and society as a whole. Ecovillagers' current lifestyles within *Travel*, *Homes* and *Diet* scopes appear to reach the decarbonisation target. Yet, ecovillagers' *Consumption* scope remains comparable to the average European citizen, causing decarbonisation targets not to be met overall. Some ecovillagers' lifestyles within the scopes are transferable, whilst others cannot be scaled up in a straightforward way to the whole European society. Hence, additional interventions for scaling up are necessary, and the bottlenecks posed by the current sociotechnical regime need to be considered and addressed for the European decarbonisation goals to be fulfilled.

This research shows that there are solutions to the climate crisis through real-life applications of sustainable lifestyle choices. More detailed research is needed to understand how the ecovillagers' lifestyles can be scaled up in Europe and what kind of policies and

interventions can support the diffusion of the most promising solutions so that they can become part of the broader sociotechnical regime.

**Supplementary Materials:** The following are available online at https://www.mdpi.com/article/10.3390/su142013611/s1, Entire Dataset Survey.

**Author Contributions:** Conceptualization, F.W., M.T.N. and M.G.G.S.; methodology, F.W.; software, F.W., M.G.G.S. and I.M.O.; validation, I.M.O. and A.K.R.; formal analysis, F.W. and M.G.G.S.; investigation, M.T.N.; resources, M.T.N.; data curation, M.T.N.; writing—original draft preparation, M.T.N., F.W. and M.G.G.S.; writing—review and editing, F.W., M.G.G.S. and M.T.N., I.M.O. and A.K.R.; visualization, F.W. and M.G.G.S.; supervision, I.M.O. and A.K.R.; project administration, F.W.; funding acquisition, I.M.O. All authors have read and agreed to the published version of the manuscript.

**Funding:** The authors acknowledge Open Access Funding by the University of Graz.

**Institutional Review Board Statement:** The paper has been internally reviewed and the submission to the journal has been internally approved.

**Informed Consent Statement:** Informed consent was obtained from all subjects involved in the study.

**Data Availability Statement:** The data are available upon request.

**Acknowledgments:** We would like to thank Michael Ortner and Jakob Stadlober for their support in the initial phase of this project. Furthermore, we would like to thank the ecovillagers that participated in the survey. Without their contribution this research would not be possible.

**Conflicts of Interest:** The authors declare no conflict of interest.

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
