# Peer review of "Scaling Up Ecovillagers’ Lifestyles Can Help to Decarbonise Europe"

_sustainability, doi:10.3390/su142013611_

Round 1

Reviewer 1 Report

General

The manuscript makes a valuable contribution in linking a bottom-up survey of personal behaviour to the energy and emission drivers of an integrated energy model. Furthermore, the authors determine that should the entirety of the EU population adopt key energy-behaviour aspects reported by ecovillagers, the resulting significant reductions in GHG emissions would help the block with achieving its climate targets. Although the result is somehow expectable, the survey-model linkage is original and the quantification of the abatement potential is an interesting metric to communicate to decision-makers and ecovillage initiatives. The authors also highlight that in some specific cases the behaviour of ecovillagers is not different from that of the average EU citizen, which can point to the existence of unsurpassable emission bottlenecks posed by geography, infrastructure or both. Unfortunately, the authors do not elaborate on this finding, in my opinion, they should. Also somehow disappointing is the utility of the Multi-Level-Prespective (MLP) framework as an integral aspect of the paper. Surely the MLP provides a neat way to map aspects of the work conducted to a framework of socio-technical transitions (as done by the authors); but beyond the aspirational transition drawn in Figure1 the framework seems to be disregarded elsewhere. This seems to be a lost opportunity because the energy model used by the authors integrates aspects of regime and landscape that could potentially be investigated. For example, the relation of ecovillagers with energy production (e.g., do they favour renewable energy, usage of bio-energy etc...) or their attitudes towards agricultural practices (e.g., intensification vs agroecology). These aspects are also inputs to the energy model used and could potentially be used to better embed the MLP into the work done. If this is not possible then there is little utility for the framework (something that is also not really discussed in the late stages of the paper) and the work done is perfectly valid without it.  The authors need to put more effort into better detailing how the answers to the survey are linked to the model and the assumptions involved. I provide some suggestions in the comments for restructuring some sections in order to reflect this. In addition, the authors should provide some lessons learned to energy modellers regarding the selection of energy drivers used as inputs to the model. For example, what major lifestyle aspects are the model missing that become apparent from the survey?. Finally, the authors apparently don't take advantage of the country-level resolution of the energy model (it allows to move the ambition lever independently for all EU26 Member states). It is not clear if this is because of the geographic heterogeneity in responses (e.g., countries with only 1 response vs countries with 20) or something else. The risk of disregarding this is that the authors might be under/overestimating the GHG reduction potential at the EU level (E.g., car occupancy is set the AL4 across EU because more than 50% replied adequately, still 14% drive with only 1 person - these lifestyles are being subsumed under AL4). The authors could test if the in-country differences in their survey are lower than the differences in answers across the full sample (for each behaviour tested). If YES then the authors have little ground not to conduct a more country-specific matching of the survey results to the levers. If NO, then authors can assume some kind of EU homogeneity across ecovillagers and keep their analysis broadly intact. In any case, this needs to be tested and communicated to the reader. Finally, the authors could discuss also a bit more how their current research and future research fit/adds to the growing focus on evaluating demand-based solutions to decarbonization (e.g, the IPCC has now a chapter dedicated to this).

Based on the above, the authors should fully attend to the suggestions/points made and return a revised version of the manuscript for further consideration.

Abstract

"Ecovillages provide examples of more sustainable lifestyles..." > this is a bit unclear. I think the authors are trying to say something along the lines "the typical behaviour of habitants in ecovillages is potential more conducive with sustainable lifestyles"

"would continue to produce significantly lower carbon per capita than the average European lifestyle by 2050" > at this point a quantification of the "significant lower" would be informative for the reader.

Name how many European countries have taken part of the survey.

Manuscript

lines 33 to 39: the authors can also add that the newest report of the IPCC reinforces the need for demand-side GHG reductions in order to comply with ambitious climate targets.

line 41: is there somehow a broad definition of what is an "ecovillage" or better said, what is the definition of "ecovillage" taken up in this study? This would be important to provide the reader with some reference framework on how to read the paper. Does not have to be an extensive definition, only some more clarity of what the term is embedded in this work.

line 53: The passage "template for sustainable living and contribute to decarbonisation pathways" is probably too ambitious. What the paper does is evaluate to what extent a generalized adoption of ecovillager's lifestyles across Europe would mean for the evolution of GHG emissions.

lines 56; consider if the authors mean "which behaviour" rather than "which lifestyle practices".

line 57: A reference to the model's web page would be appropriate at this point (https://www.european-calculator.eu/model/).

line 84: ok... this is indeed some type of definition and would be nice to incorporate a short version as a reply to a comment made in line 53.

line 91: the authors mean resisted in the Global Ecovillage Network right?

line 92: is there any estimate of the total amount of European population living in the registered ecovillages?

lines 95 to 119: the introduction of the MLP of systems innovation is rather interesting. So far into the paper, the authors seem more to translate the transition narrative from Gells into the specific case of ecovillagers > decarbonization. Nothing wrong with that but the linkage remains qualitative and not formal. Curious how the authors will link the MLP to the formal aspect of the EUCalc.

line 123: the model used by the authors has been better described in the publication below rather than in the policy paper currently cited in the manuscript.

Luís Costa, Vincent Moreau, Boris Thurm, Wusheng Yu, Francesco Clora, Gino Baudry, Hannes Warmuth, Bernd Hezel, Tobias Seydewitz, Ana Ranković, Garret Kelly and Jürgen P. Kropp. The decarbonisation of Europe powered by lifestyle changes, Environmental Research Letters, 2021.  

line 130: the authors are right that optimization models are often the norm in the low-carbon analysis" but EUCalc is not an optimization model. It's advisable the authors have a look at the publication above.

line 147: probably more correctly is to say that "the focus is on the energy and emission drivers under Key behaviours of the EUCalc model".

line 151: "Values for necessary adjustments for each indicator" > not clear to me what the authors are trying to say here.

line 152: the authors seem to be referring to a survey they created with questions based on the EUCalc interface and lever definition. In combination with the comment in line 151 I would advise the authors to spend some more time showing the reader a) some of the questions elaborated, and b) how the answers given by the villagers were recorded. In addition, a few more key figures regarding how many ecovillages were surveyed, their geographic distribution etc would be commendable. Is a nutshell... a bit more explanation on the survey set-up and survey process would be required. In relation to this, what is currently Figure 3 could be done to portray a general case instead of the specific option of car occupancy. By contrast, it would be advisable that in figure 2 the full list of divers under investigation is provided, could be a table...

line 162: As suggestion, I would rename what is currently section "3.2. Data Collection" to "Survey process" in which I would explain in subsections the a) construction of the survey from the key behaviour levers of the EUCalc, b) how the information was collected, and c) details of the sample size, geography etc... that would mean also to incorporative what is currently section "3.2. Data Collection"... 

line 200: I have trouble reading from figure 5 what part is a descriptive analysis and what consists of an explorative analysis. I might be wrong but the authors seem to show some type of "aggregation" of answers or mapping of answers to the levers of the EUCalc. This figure needs some more explanation or even re-thinking in my opinion. Why is it that for the cases in which "Not meeting AL4" figure 5 shows what seems to be AL3 in the EUCalc?

line 204: it is still unknown to the reader what the questions look like, very hard to judge.

line 205: this points out for the authors having a higher granularity of questions that they subsequently aggregate. It would be great to have a much better idea of what the full list of questions was.

line 211: the EUCalc works on a country resolution! In order to be consistent, the authors should match their country-level survey to country-level adjustment of ambition levers. Leaving things at the EU-level results in GHG savings that can be under/overestimated because they do not account for the heterogeneity in survey responses across countries.

line 213: "The explorative analysis aimed to identify the potential for widespread adoption..." > the authors need to better explain the rationale behind this statement.

line 217: "Indicators meeting AL4 are set to AL4 and those not meeting AL4 remain with the current EU reference value..." This should be reflected in Fig.5. In this part of the text, the authors seem to be using "EU reference value" as "European average" shown in Fig.5. Try to keep the terminology consistent. In addition 

lines 217 to 224: it seems that the authors are generating several scenarios with the EUcalc that reflect the survey answers and others are more hypothetical. If this is the case it would be advisable to either name them clearly at this point.

line 218: "EU reference value (pre-set by EUCalc)"... a bit more detail would be good. Looking into the EUCal website the value that the authors are referring to "The combination of lever positions under this scenario reproduces, as far as possible, the main sectoral assumptions and outputs of the EU-Reference scenario as detailed in Capros et al 2016. Capros P, De Vita A, Tasios N, Siskos P, Kannavou M, Petropoulos A, Evangelopoulou S, Zampara M, et al. (2016)."

line 240: From my perspective, the examples in table 2 would be really useful earlier in the paper in explaining how the EUCalc and survey were linked. In fact, if one excludes the % results section "4.1. Ambition levels (Als) of lifestyle levers in ecovillages" describes the methodological choices the authors faced in order to link the EUCalc and the survey. As mentioned before, I would see this as part of the methods and less as part of the results. In the results, I would simply report the findings of what key behavioural of the ecovillagers matched the AL4 and which did not - for all questions in which this was feasible.

Table 2: Is there a reason why the colour scheme of the pie charts in "Appliances owned" is different from the others?

line 282: The passage "which lifestyle aspects of eco-living can be considered relevant and transferable to broader society." is interesting but one would expect that the authors say a bit more about what measure of "relevance" and "transferability" they are considering. Is it meant only in terms of abatement potential? or something more?

line 294: for the sake of completeness mention that indicators in grey are those that were hard to survey.

line 305: provide also a brief note on the assumptions made for the levers that are not the focus of the exercise. One assumes that they are left unchanged at EU reference but better mention this explicit.

line 305: I understand the value statement behind labelling level 4 as "desired lifestyle" but within a research paper less nuanced language is preferable. Accordingly, I would suggest the authors frame level 4 not as "desired" but as the "behaviour with maximum GHG abatement potential considered in the EUCalc". It is less appealing but more correct...

line 309: "when adopted by the broader society", probably would be more correct to say "if adopted"

Figure 11: It might be worth it for authors to explore the amount of negative emissions that are impacted by the ecovillagers lifestyles. Negative emissions in Europe are likely to be potentiated given the less consumption of meat, this would help in the decarbonization challenge and also have some benefits in your carbon budget calculation.

line 382 to 393: check the adequacy of citation [18] in supporting the statements made, it might be better to provide some more papers. Another issue that the authors might explore is the geographic location of ecovillages. As population densities decrease (e.g., in rural areas) one has to travel on average further to get access to services. Accordingly, the ecovillagers could be locking themselves in travelling higher distances simply due to geography. This means that even what is broadly a more sustainable way of life comes with its own trade-offs, highlighting how challenging decarbonization really is. From the reading of the discussion in lines 436-447, the same trade-off appears to emerge in living space. The authors are advised to explore this nexus at probably hinting at some technological developments that would lower such trade-offs, for example, more remote working and online services could avoid unnecessary travelling, clever energy retrofitting and automate-systems could offset some of the GHG that come with living, unavoidably??, in bigger houses?

line 474: in addition, the authors can also highlight that some key behaviours that can be quite environmentally damaging are not accounted for explicitly, for example, clothing and furniture purchases. Actually, from my perspective, it would be useful to communicate what the modellers can learn from the survey in terms of specific behaviours from ecovillagers that are not accounted for in the EUCalculator. Spending some time on this could be very informative, for example, behaviour around the circular economy or the relation of ecovillagers to time spent on certain activities. It could be very relevant for emissions to investigate the relation of ecovillagers with energy production and land practices, for example, is there a disproportional adoption of green energy providers; what agricultural practices do they favour. Note that also these "levers" can be changed in the EUCalc and although they might not be a behavioural decision in the most strict sense they could be subsumed as part of the ecovillager "lifestyle". I also understand that such analysis might not be possible at this point as the survey made had a very clear focus in mind. If this is the case then the authors should at least point the reader to understand that there are other related energy issues (some of them at the level of regime or landscape, see below) that the ecovillagers lifestyles relate to and are not investigated in the paper.  

line 488 to 490: So far the inclusion of the MLP in this work is a bit disappointing. Cannot grasp why one needs it at all. In my view, the paper is perfectly fine without it and its main findings do not require an MLP in order to be sound. From what I know the MLP is a description of how socio-technical transitions can be stylized along three levels, but in this paper - and from what I'm concerned - the authors do not really leave the level of "niche". Although the authors apply a model that is broader in scale its use is from my understanding just a tool to "upscale" the potential benefits of ecovillagers lifestyles. The authors do not venture into expanding via which process the "niche" lifestyles influence the average European lifestyle, they simply investigate what are the GHG implication if all EU citizens live in the same manner as their survey sample regarding particular behaviours. My understanding of the MLP is to expose how could this take place and via which mechanisms, policies etc... That said the authors should consider a better placement of the MLP in their manuscript highlighting why is it important for the main message of the paper.

line 527: It is rather puzzling that authors refer that ecovillagers lifestyles cannot "act as the sole niche innovation toward decarbonisation". It seems to contradict to a large extent Figure 1 makes even more clear the previous comment needs to be addressed in depth by the authors regarding the utility/role of the MLP.

Author Response

Dear reviewer,

Thank you for the feedback on our article. Please find the responses to your comments in the document attached below.

Otherwise we will be happy to resubmit it.

Kind regards,

Franziska Maria Wiest, Gabriela Gamarra Scavone , Maya Tsuboya Newell , Ilona M. Otto , Andrew K. Ringsmuth

Reviewer 2 Report

What an interesting combination of empirical research and thought experiment. Many of us who have studied ecovillages have wondered: what if everyone lived like the people we are studying? And I like that someone has taken steps to start answering that question.

This paper needs an added element of theoretical grounding. Why is decarbonization an important goal, and why are other attempts at it inferior to the cultural experimentation offered at ecovillages? Alternatively, why are ecovillages interesting as an object of study to the overall agenda of decarbonization? I have some suggestions to this effect, most notably that Bourdieu’s concept of Habitus has been applied to ecovillages and that provides are ready framework for relevance to theory. See, for example:

Haluza-DeLay, Randolph, and Ron Berezan. 2013. “Permaculture in the City: Ecological Habitus and the Distributed Ecovillage.” in Environmental Anthropology Engaging Ectopia, edited by J. Lockyer and J. T. Veteto. New York: Bergehahn Books.

Kasper, Debbie VS. 2009. “Ecological Habitus: Toward a Better Understanding of Socioecological Relations.” Organization & Environment 22(3):311–26.

There are other theories which could be adapted in service to forwarding the use of MLP as an appropriate method which have not yet been used in the study of ecovillages. For example, James Coleman or Raymond Boundon’s theories of social action.

It also needs an added group of literature addressing the question of whether people are likely to adopt the behaviors of ecovillagers. As much as answer the question “what if everyone lived this way?” is exciting, it needs to also be paired with consideration of “how likely are people to live this way?” to assess the relevance of findings. There is a hint of this section 5 but should also be in the literature review and bridged to the development of a richer theoretical grounding.

A major strength of this paper is in the visuals. Tables 2 and 3, as well as Figures 1, 10, 11, and 13 (as currently numbered). Those add a great capacity for explaining the results.

Additional commentary by line:

Line 46: starts abruptly with no subject. Who reviewed 59 studies?

Line 48: Similarly, no subject offered.

Line 110: No subject in this sentence.

Line 168: “Ecovillages were contacted and invited to participate…” How? This section should contained more details about how the sample was reached, including appropriately labeling it variously as snowball and convenience sampling.  

Line 300: The progression of figures skips from 5 above to 10 here. Adjust this and all subsequent figures to be in the correct order.

Line 323: “CO2eq” seems to be a typo.

Line 426: Delete “concluding”

Line 518: “Europe-wide” not “Europe wide”

Normally, the omission of subjects would suggest to me that authors are blinding self-citations for the purpose of review, but that does not seem to be the case here – especially as the authors are not blinded from the reviewer.

Additional commentary by section:

Section 2.1

What are ecovillages? I don’t mean this to question the authors’ expertise that they clearly have, but needs to be set out at the beginning to establish a point of common connection with the reader. The best way to address this is probably to define the concept based on the group captured in the data collection process.

Section 2.2

This section needs to be developed further to show how MLP has been used in other studies, and how these illustrate the merits of using it.

Section 3.2

There needs to be more detail about how the data was collected. Most notably, the authors report 73 residents of ecovillages responding but not how many unique ecovillages were represented. What was done in cases where more than one person from a particular ecovillage responded? That responses were coming from household-level samples should be noted and discussed in detail in the limitations. This article (pages 5-6) also deals with the same problems:

Rubin, Zach, Don Willis, and Yana Ludwig. 2019. “Measuring Success in Intentional Communities: A Critical Evaluation of Commitment and Longevity Theories.” Sociological Spectrum 39(3):181–93.

It doesn’t add much to note the attributes of incomplete surveys. Unless the authors can point to a common reason they weren’t completed, the number of views and starts isn’t relevant.

Sections 4 and 5.

The discussion and conclusion are very well organized and articulated.

Author Response

Dear Reviewer,

Thank you for the feedback on our article. Please find the responses to your comments in the document attached below.

Best wishes,

Franziska Maria Wiest, Gabriela Gamarra Scavone , Maya Tsuboya Newell , Ilona M. Otto , Andrew K. Ringsmuth

Round 2

Reviewer 1 Report

I thank the authors for taking on board the majority of my suggestions as well as providing an answer for all clarification requests made.  I have the opinion that the manuscript increased in both clarity and readability.

That said, I have the opinion that a few minor points (listed below) still need to be resolved and the final manuscript is subject to final proofreading in order to sharpen up the English and clear up some inaccuracies. 

I suggest the opening of section 3.1 to be: "The EUCalc models energy, resources, production and food systems at the EU level under pre-defined (but adjustable) levels of ambitions in regard to technological deployment and consumption behaviour."

line 300: "Based on the example of Figure 3, Table 3 shows the process of creating the survey question and answer options based on the AL4 threshold obtained from EUCalc (Table 3)." I suggest this text be changed to: "Table 3 shows how answers from the survey are used to define whether a lever should be set to AL4."

line 321: For this table and throughout the paper, the authors should harmonize the use of "lever" and "indicator" as follows: Travel, Homes, Diet and Consumption are "scopes"; "Occupancy", "Type of diet", "Appliances use" are the levers; and the drivers of energy defining the ambitions of each lever are what you call "indicators".

line 321: There is no caption mentioning the number of this table. The authors need to revise the numbering and positioning of the tables and figures throughout the document.

line 397: "If less than 50% of the respondents indicated an answer complying with “Meeting AL4”, the respective indicator was set to the European reference." The authors should replace this passage with: "If less than 50% of the respondents indicated an answer complying with “Meeting AL4”, the respective indicator was set to the level that is better aligned with the European reference scenario in [27]."

line: 452: In the table "Overview of relevant survey results (source: own survey data)" authors should keep consistent the colouring of the shares in each pie chart: "meeting AL4" - green, "not meeting AL4" - orange.

line 543: In figures 5, 8,9 and throughout the text the authors need to replace "European average lifestyles" with "European lifestyle aligned with [27]" every time that ambition level is not set to AL4. Furthermore, the figure contains some portion of the negative y-axis although nothing is said in the caption about negative emissions. The authors should revise the plots in figures 8 and 9 

line 826. In the first round of review, the authors were asked to specify "what major lifestyle aspects are the model missing that become apparent from the survey?" At the end of section 5.2, the authors provide some general citations but the request was made specifically in the context of the work being carried out, not regarding previous work. I urge the authors to provide some further insights regarding what can energy modellers learn from their study in order to improve the models.

Reviewer 2 Report

This draft is much improved, and I think a very valuable contribution to the literature on how ecovillages can be demonstrable examples of a more sustainable world. I look forward citing the top-line result in my own work someday soon, that if everyone lived like an ecovillager in Europe it would lead to a 33.3% reduction in CO2 emissions but that would still exceed the hypothetical budget to avoid 1.5C warming.

I very much appreciated the increased discussion of the role of carbon and decarbonization throughout the literature review and methods. That seems to be in response to another reviewer, but it improved the paper in a way that made wish I had thought to give that feedback. The overall explanation of the role that AL4 and the relative pathways to decarbonization between ecovillagers and the “average” European has been rendered much more clear in this draft. 

This also satisfies what I described in the previous review as a lack of theoretical grounding. I can understand the authors’ response that the theories I suggested require more of a cultural grounding in ecovillage lifestyles, and so a more macro approach was warranted here. I did not mean to suggest that famous people or their names were necessary, only a starting point we can assume about the world. That we need to move urgently towards decarbonization – not just the EU but the world as a whole – suffices here. 

Just a few quick edits:

Line 72: “ecovillagers” not “ecovillager,” per context.

Line 82: “This paper aims to what extent analyse ecovillagers’ lifestyles…” is grammatically awkward.

Line 385: “Hereby” is an awkward word choice. Maybe “Hence”?

Line 640: “within” not “withing”
